# Digital Mapping of Soil Properties Using Ensemble Machine Learning Approaches in an Agricultural Lowland Area of Lombardy, Italy

Odunayo David Adeniyi [1] , Alexander Brenning [2] , Alice Bernini [1] , Stefano Brenna [3] and Michael Maerker [1,4,*]

1 Department of Earth and Environmental Sciences, University of Pavia, 27100 Pavia, Italy
2 Department of Geography, Friedrich Schiller University Jena, 07743 Jena, Germany
3 ERSAF, Regione Lombardia Milan, 20124 Milano, Italy
4 Leibniz Centre for Agricultural Landscape Research, Working Group on Soil Erosion and Feedbacks, 15374 Müncheberg, Germany
* Correspondence: michael.maerker@zalf.de or michael.maerker@unipv.it; Tel.: +49-176-6266-3563

**Abstract:** Sustainable agricultural landscape management needs reliable and accurate soil maps and updated geospatial soil information. Recently, machine learning (ML) models have commonly been used in digital soil mapping, together with limited data, for various types of landscapes. In this study, we tested linear and nonlinear ML models in predicting and mapping soil properties in an agricultural lowland landscape of Lombardy region, Italy. We further evaluated the ability of an ensemble learning model, based on a stacking approach, to predict the spatial variation of soil properties, such as sand, silt, and clay contents, soil organic carbon content, pH, and topsoil depth. Therefore, we combined the predictions of the base learners (ML models) with two meta-learners. Prediction accuracies were assessed using a nested cross-validation procedure. Nonetheless, the nonlinear single models generally performed well, with RF having the best results; the stacking models did not outperform all the individual base learners. The most important topographic predictors of the soil properties were vertical distance to channel network and channel network base level. The results yield valuable information for sustainable land use in an area with a particular soil water cycle, as well as for future climate and socioeconomic changes influencing water content, soil pollution dynamics, and food security.

**Keywords:** digital soil mapping; ensemble machine learning; stacking model; terrain attributes; Lombardy lowland

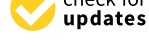



## 1. Introduction

The soil is the most crucial part of our ecosystem and its functioning in terms of crop production, filtering of water, hosting and maintaining soil biodiversity, atmospheric carbon sequestration and storage, as well as biomass production. Soil functions, in turn, depend on soil properties, such as water holding capacity, soil available nutrients, soil organic carbon stock, etc., that can be portrayed by soil maps [1]. Today, precise soil information with high spatial resolution is in great demand by various stakeholders, including soil scientists, land use planners, environmental managers, and farmland managers. Traditional soil surveys manually delineate discrete, vector-type soil units that are difficult to update since there is a need to repeat the entire production procedure that, in part, is subjective and based on expert knowledge [2]. This traditional method also requires numerous soil samples, and it is therefore expensive and time-consuming. Even though classical soil surveys are a fundamental prerequisite for digital soil mapping (DSM), the latter allows for the overcoming of some limitations of the classical methods using available, spatially distributed auxiliary environmental information and Geographical Information Systems (GIS).

Generally, DSM estimates the properties of soil by analyzing the relationships between soil characteristics and the environmental variables, using geostatistical and machine learning (ML) models [3,4]. The available environmental variables play an important role in predicting soil properties across different landscapes, especially in complex terrain. Soil scientists identify topography as one of the main pedogenic factors, which significantly influences the spatial distribution of soil properties (e.g., [5]). Studies like Grimm et al. [6], Seibert et al. [7], Tu et al. [8], or Song et al. [9] showed that exclusively using terrain attributes yields the potential to effectively map the spatial distribution of soil properties. However, most agricultural lowland areas often show weak correlations between the input variables and specific soil properties [10,11]. These low performances in lowland areas are due to the landscape being characterized by a low-gradient relief, and thus, an accurate prediction of soil properties is quite challenging. To tackle this challenge, different modelling approaches are generally compared to choose a single 'best' model or an 'optimal' set of models to improve prediction accuracy by reducing the uncertainties of predicted values.

The advantage of ML algorithms is related to the ability to quantify the high-dimensional and nonlinear relationships between soil properties and environmental variables over diverse soil landscapes [12]. The application of ML techniques in DSM helps to improve the prediction of soil properties, hereby overcoming some of the limitations of conventional soil mapping approaches [13,14]. ML is also suitable in DSM if data availability is limited [15]. Several studies have applied novel ML techniques in DSM to predict the spatial distribution of soil properties and types [12,16,17]. Some of the most common ML models used in DSM are support vector machines, multivariate regressions, regression trees, Cubist, random forest, and gradient boosting machines [18–20]. The emergence of different ML models has encouraged model comparison studies in which different models might generate distinctly different digital soil maps, despite using the same input data [12,14,16]. As a result of this, it is advisable, for the best practice in DSM, to compare and evaluate different model techniques [12] and choose the best performing one [16,21]. However, selecting the best performing model could be problematic because each model has its own pros and cons in specific circumstances. Thus, one model could perform better than others in a certain situation and area [22,23]. Therefore, another approach that helps to combine the information and knowledge acquired from single models is ensemble modelling [24,25]. Ensemble models result in potentially better and more stable predictions, in comparison to predictions made using single ML models. Moreover, they reduce the risk of choosing the "wrong" model [26,27]. Random forest, which applies a bagging method, and gradient boosting machines are common ensemble learning ML algorithms that are used in DSM [28]. However, these ensemble models were built using a single type of predictive learner (homogenous ensemble learning), and less attention has been paid to modelling approaches that combine multiple types of ML models as base learners (heterogenous ensemble learning) within DSM studies. Model averaging is another ensemble technique that was proposed [29–32].

Stacked generalization is a type of ensemble learning and model averaging approach. It involves training a new learning algorithm to combine the predictions of several base learners. Several trained base learners are aggregated into a combined learner using a combiner algorithm called the 'meta-learner'. The latter is based on the hypothesis that the combined model has a better predictive performance [33,34]. Here, the meta-learner evaluates the predictive performance of the individual base learners and builds an optimal combination [35]. This approach accounts for the differences in the predictive performance of the base learners [36]. Unlike other ensemble models, the stacking approach has rarely been explored in DSM; nevertheless, this approach often out-performs individual models [37,38].

Ensemble learning with stacked generalization combines the results from multiple ML algorithms to further develop an integrated mapping output, with relatively stable performance. To the knowledge of the authors, this approach is relatively uncommon in DSM, especially for lowland areas. First attempts were presented by Taghizadeh-Mehrjardi et al. [37,38]

who used a stacked generalization of ensemble ML models to predict SOC content, and a super learner for other soil properties; Zhang et al. [39] also used this approach to predict soil pH. Hence, the objective of this study is to evaluate and compare a stacking ensemble model approach with five ML models (base learners) to predict and map the spatial distribution of different soil properties, such as texture (sand, silt, clay content), soil organic carbon (SOC), pH, and topsoil depth, in an agricultural lowland area of Lombardy region, Italy. Diagnostic tools for the interpretation of these black-box models were applied to assess their plausibility, as well as similarities and differences, in the modelled relationships, which reflect the related model's abilities and biases.

## 2. Materials and Methods

### 2.1. Study Area

The study area (Figure 1) covers approximately 314 km² and is located about 15 km southwest of the city of Milan, in the Lombardy region, close to the border with the Piedmont region. The area is part of the Ticino River valley and the elevation ranges between 64 m.a.s.l, in the southern part of the Ticino River, to 135 m in the northern parts (Figure 2). The Ticino River is the only natural drainage system in the investigated region. The area, in fact, is characterized by a strong anthropogenic influence and is constantly evolving. The area is intensively cultivated, and the main crops are maize and rice, irrigated through artificial canals. The land use and land management practices date back to the eleventh century with the construction of irrigation channels [40] and the reuse of water along the fluvial terrace cascade of the Ticino River, representing, for centuries, an example of a sustainable and effective reuse of irrigation water.

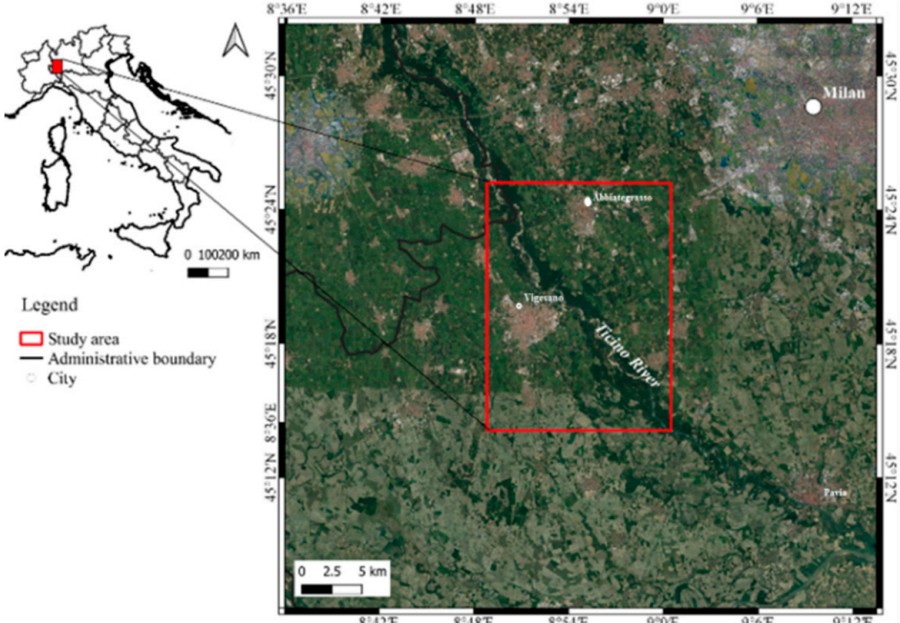

**Figure 1.** General overview of Italy and focus on the study area between Abbiategrasso and Vigevano in Pavia Province.

The area is mainly flat, except for the river terraces that have been incised by the Ticino River, generating escarpments with maximum inclinations of 30 degrees. The soil shows a sandy loam texture, developed on Quaternary alluvial deposits. Particularly, the area is characterized by Pleistocene fluvial and fluvio-glacial, gravelly to sandy sediments deposited in the last (i.e., Würm) glaciation, as well as more recent Holocene fluvial deposits, with a mainly sandy-gravelly and slightly silty character.

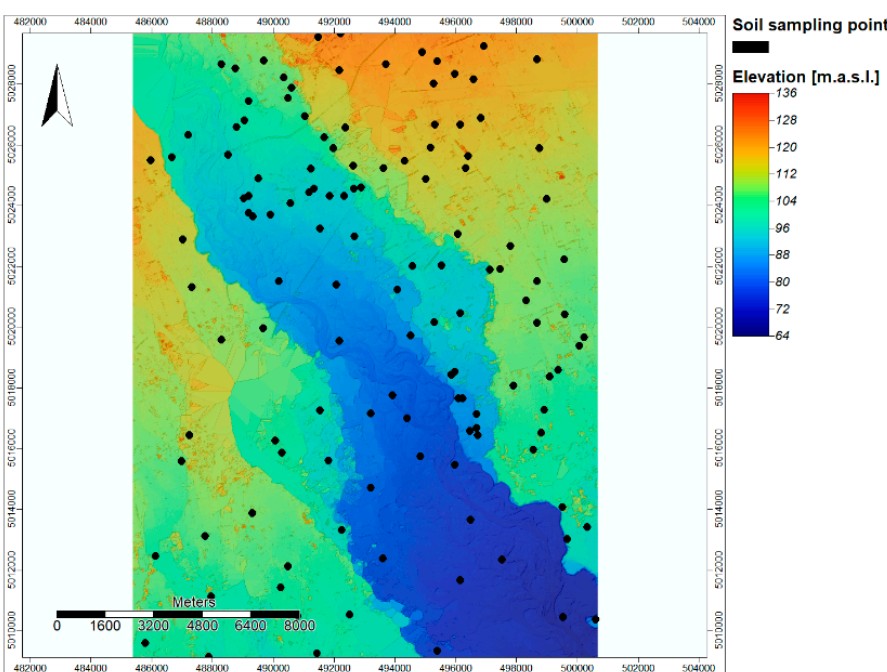

**Figure 2.** Hybrid digital elevation model with 10 m resolution based on TanDEM-X (12 m resolution) and Lidar (1 m) digital terrain models. Color-coded elevation with hill shading. Black dots show the location of the sampled soil profiles.

The region has a humid subtropical climate (Cfa), following the Köppen climate classification [41], with warm summers and cold winters.

Soil profile data (n = 130) was provided by ERSAF (Ente Regionale per i Servizi all'Agricoltura e dalle Foreste) [42] and described specific soil properties, such as soil pH in water, soil organic carbon (SOC%), texture (sand, silt, clay content in %), and topsoil depth (cm). Generally, the soils are characterized by a sandy loam texture developed on Quaternary alluvial deposits.

In this study, we modelled the soil properties texture (sand, silt, clay content), soil organic carbon (SOC), pH, and topsoil depth by using multiple base learners, and compared them against an ensemble learning approach with stacked generalization. The performances of this approach were compared with the base learners, and the best model was used to develop the digital soil maps.

## 2.2. Environmental Variables

The environmental conditions were represented by terrain attributes, land use, and landcover maps (LULC). In this study, LULC is used to represent the influence of human activities on soil properties distribution. The LULC map, for the year 2018, was obtained from the geoportal of the Lombardy region (https://www.geoportale.regione.lombardia.it, accessed on 1 February 2023). These maps were produced using SPOT6/7 2018 satellite image and had a spatial resolution of 1.5 m. The provided land cover types were reorganized into simple arable land, rice fields, and broad-leaved forest, with medium and high density governed by coppice (Figure 3).

Terrain attributes are the most extensively used environmental variables in DSM [43]. They are proxies for solute, water, and sediment fluxes through the landscape. In this study, the terrain attributes were derived from a 10 m resolution hybrid digital elevation model, obtained from the interpolation of a TanDEM-X DEM with 12 m resolution (provided by Deutsches Zentrum für Luft- und Raumfahrt, DLR) [44] and a 1 m resolution Lidar digital terrain model (DTM), acquired from the Ministry of the Environment and the Protection of the Territory and the Sea [45]. The DEM was pre-processed by filling gaps and removing artefacts following Maerker et al. [46]. Subsequently, the terrain attributes representing the

environmental conditions include topographic wetness index (TWI), multi-resolution ridge top flatness index (MRRTF), multi-resolution index of valley bottom flatness (MRVBF), modified catchment area (MCA), mid-slope position (MSP), slope height (SH), channel network base level (CNBL), and vertical distance to channel network (VDCN). McKenzie et al. [47] discussed the role of terrain analysis in soil mapping. These topographic indices were extracted from the pre-processed DEM using the System for Automated Geoscientific Analysis (SAGA) software (version 8.2) [48].

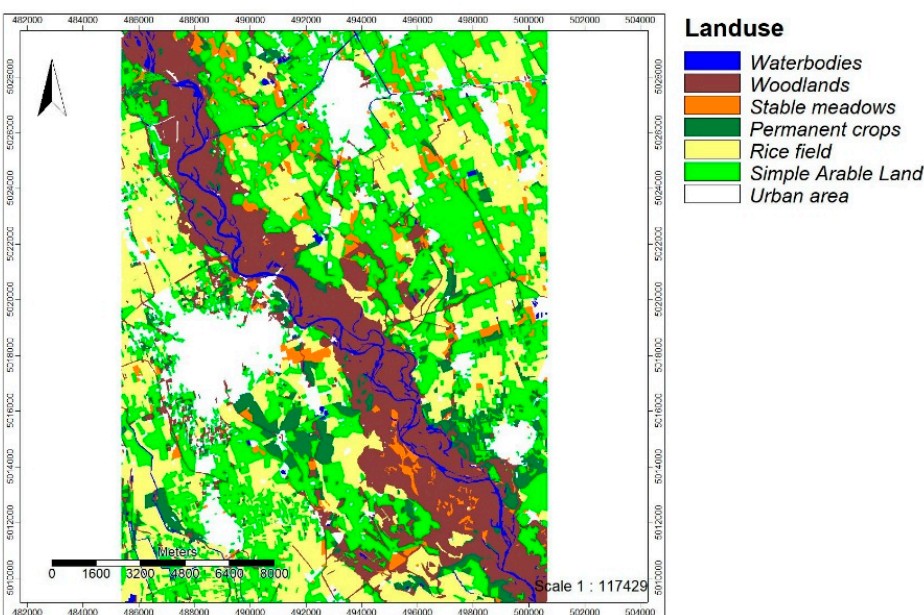

**Figure 3.** Land use and land cover map for the year 2018 (source: geoportale Lombardia).

*2.3. Base Learners*

Five ML models (Table 1) were used to identify the relationships between different soil properties and environmental variables for our study area. These models included Cubist, gradient boosting machine (GBM), generalized linear model (GLM), random forest (RF), and support vector machines (SVM). RF [49] and GBM [50] are homogenous ensemble models which consist of a non-parametric technique that combines predictions made by multiple decision trees.

RF is based on a bagging algorithm. It uses the bootstrap strategy to resample observations, and it randomly selects a subset of the features to build an ensemble of regression trees, whose predictions are averaged. Hereby, it effectively reduces the problem of overfitting each model. The RF prediction is performed using the "rf" function in the "caret" package in R. GBM, instead, uses a boosting algorithm, which gradually builds a tree-based model by fitting additional learners to the errors of the model built up to that point. In this study, GBM was modeled by the "gbm" function of the "caret" package. Cubist is an advanced regression tree algorithm [51] that combines decision trees and multiple linear regression methods and adds multiple training committees and boosting to make the weights of the trees more balanced. In this study, the "Cubist" package and the "caret" package were combined for regression modeling.

SVMs are a popular supervised learning technique for classification and regression that are capable of modelling nonlinear relationships that can be generalized to nonlinear models using kernel functions, as proposed by Cortes [52]. The radial basis function (RBF) kernel, which has been widely used in soil mapping research [53–55], was selected as the kernel of the SVM algorithm. In this study, SVM was modeled by the "svmRadial" function of the "caret" package. GLM is a linear regression algorithm which uses the ordinary-least-squares method to determine the coefficients of its independent variables and the intercept value by minimizing the sum of squared residuals. In this study, GLM was modeled by the

"glm" function of the "caret" package. All the hyperparameters for each model (Table 1) were tuned with internal cross-validation, i.e., by performing an 'inner' cross-validation on the training set without looking at the test sample used for model assessment [56].

**Table 1.** List of models and corresponding hyperparameters in caret.

| Base Learners | | Hyperparameters | Grid Search | Reference |
|---|---|---|---|---|
| Cubist | Cubist | committees neighbours | 5 to 50 (step size 5) 1, 5, 9 | [57] |
| Stochastic Gradient Boosting | GBM | n.trees | 100 to 800 (step size 50) | [50] |
| | | interaction.depth Shrinkage minobsinnode | 1, 3, 5, 5, 7 0.001 to 0.01 10, 15, 20 | |
| Generalized Linear Model | GLM | None | | [58] |
| Random Forest | RF | mtry | 2 to 15 | [49] |
| Support Vector Machine | SVM | $\sigma$ | $10^{-5}$ to $10^3$ (length = 15) | [59] |
| | | C | $10^{-5}$ to $10^3$ (length = 15) | |

### 2.4. Stacking Generalization

The ensemble machine learning approach, known as stacking generalization, was employed to combine the individual ML model predictions (as base learners) and to maximize the generalization accuracy. The predictions of the five base learners were combined using a meta-learning model. Stacking helps to explore the solution space with different models in the same study. In this study, two stacking ensemble learning models were compared, as a simple meta-learner, to stack the five base learners using the "caretStack" function in the "caretEnsemble" packages in R 3.5.2 [60]. The first was a GLM model (Stack_GLM), which uses a linear model to calculate the weighted sum of the predictions made by the base learners. The second was a GBM model (Stack_GBM), which deals with non-linear trends and provides great predictive performance.

The ensemble machine learning modelling is a black-box algorithm, which poses the challenge of quantifying and evaluating the exact contributions of the predictors to the final model output. Model-agnostic interpretation tools help in handling this challenge, which may be used for any ML model. Model-agnostic methods operate by changing the inputs of the ML model and measuring the corresponding changes in the prediction output. In this study, variable importance was estimated for the five base learners using the permutation method, which is implemented in the iml package in R [61].

### 2.5. Model Prediction Performance Assessment

The model performances were evaluated using a cross validation method, as it is beneficial for small datasets, detects overfitting, and provides error estimates with comparatively good bias and variance properties [62,63]. The cross-validation approach provides a structure for constructing several training/test sets from the dataset, guaranteeing that each data point is part of the test set at least once. A nested cross-validation was applied to build and test the base learners and the ensemble models [56]. Ten-fold cross validation, with 20 repetitions, was applied to optimize the model settings (hyperparameter tuning) and to validate the final performance of the base learners, built on optimized settings. The prediction performance of all models was examined using the root mean square error (RMSE) and Lin's concordance correlation coefficient (CCC):

$$RMSE = \sqrt{\frac{1}{n}\sum_{i=1}^{n}(x\_actual - x\_predicted)^2} \tag{1}$$

$$CCC = \frac{2r\sigma_{actual}}{\sigma_{actual}^2 + \sigma_{predicted}^2 + \left(\bar{x}\_actual - \bar{x}\_predicted\right)^2} \qquad (2)$$

where n is the number of soil samples; $x\_predicted$ is the predicted value derived by each model; $x\_actual$ is the actual soil property value; $\bar{x}_{actual}$ and $\bar{x}_{predicted}$ are the averages of actual and predicted values, respectively; $\sigma_{actual}$ and $\sigma_{predicted}$ are the corresponding standard deviations; and $r$ is the correlation coefficient of the predicted and actual values. These validation criteria were used to evaluate and choose the best-performing models. While the RMSE has the advantage of measuring the prediction error in the original units of the predicted variable, the CCC provides a measure of agreement between predictions and observations. Both indicators account for both bias and random variability.

## 3. Results

### 3.1. Descriptive Summary of Soil Properties

A summary of the different soil properties in the study area is presented in Table 2. The soil sand, silt, and clay contents in the study area varied from 37.0 to 98.6%, 0.30 to 49.10%, and 1.0 to 17.30%, respectively. SOC varied from 0.50 to 4.70 g/kg, pH from 4.40 to 7.80, and topsoil depth from 4 to 62.0 cm. The pH had the lowest coefficient of variation (CV = 10.32%), followed by sand content, depth of topsoil, silt content, clay content, and SOC content (CV = 18.36, 34.04, 39.33, 63.91, and 52.73%, respectively). The skewness value of SOC shows that the statistical distribution of SOC values is skewed to the right (skewness = 1.11). Therefore, a transformation with the natural logarithm was used to obtain a more symmetric SOC data distribution. The transformed data was used for the modelling, and the predicted values from the model outputs were back transformed before accessing the model performance.

**Table 2.** Descriptive statistical summary of soil properties in the study area. Qi: i-th percentile; SD: standard deviation; CV: coefficient of variation.

| Soil Property | Minimum | Maximum | Mean | Q$_{25}$ | Q$_{50}$ | Q$_{75}$ | SD | CV (%) | Skewness |
|---|---|---|---|---|---|---|---|---|---|
| Sand (%) | 37.0 | 98.6 | 67.92 | 59.20 | 69.75 | 76.22 | 12.46 | 18.36 | −0.32 |
| Silt (%) | 0.30 | 49.10 | 26.01 | 18.07 | 25.55 | 32.85 | 10.23 | 39.33 | 0.25 |
| Clay (%) | 1.00 | 17.30 | 5.15 | 3.05 | 5.15 | 8.90 | 3.89 | 63.91 | 0.77 |
| SOC (g/kg) | 0.50 | 4.70 | 1.65 | 1.01 | 1.46 | 1.89 | 0.87 | 52.73 | 1.11 |
| log(SOC) | −0.69 | 1.55 | 0.38 | 0.01 | 0.38 | 0.64 | 0.48 | 128 | 0.31 |
| pH | 4.40 | 7.80 | 6.11 | 5.70 | 6.10 | 6.60 | 0.63 | 10.32 | 0.02 |
| Topsoil depth (cm) | 4.0 | 62.0 | 31.67 | 25.0 | 31.18 | 40.0 | 10.78 | 34.04 | −0.65 |

The predictors were not strongly correlated to each other (Figure 4). The vertical distance to channel network (VDCN) is significantly correlated with all the soil properties, and pH is, in turn, significantly correlated with the channel network base level (CNBL) (Table 3).

**Table 3.** Spearman's rank correlation rho between soil properties and terrain attributes.

| | Topsoil Depth | Sand | Silt | Clay | pH | SOC |
|---|---|---|---|---|---|---|
| CNBL | −0.03 | −0.21 * | 0.25 ** | −0.10 | 0.30 ** | 0.35 *** |
| MCA | 0.04 | −0.11 | 0.04 | 0.21 * | −0.09 | −0.09 |
| MRRTF | −0.01 | −0.20 * | 0.19 * | 0.15 | 0.03 | −0.04 |
| MRVBF | −0.30 ** | 0.05 | −0.03 | −0.09 | −0.16 * | 0.35 *** |
| SH | 0.19 * | 0.18 * | −0.22 * | 0.07 | 0.03 | −0.12 |
| TPI | −0.08 | −0.04 | −0.01 | −0.11 | 0.01 | 0.004 |
| TWI | −0.01 | −0.10 | −0.05 | 0.16 | −0.19 * | −0.07 |
| VDCN | 0.23 ** | −0.25 ** | 0.23 * | 0.28 ** | 0.02 | −0.46 *** |

* Correlation is significant at $\alpha = 0.05$; ** Correlation is significant at $\alpha = 0.005$; *** Correlation is significant at $\alpha = 0.0001$.

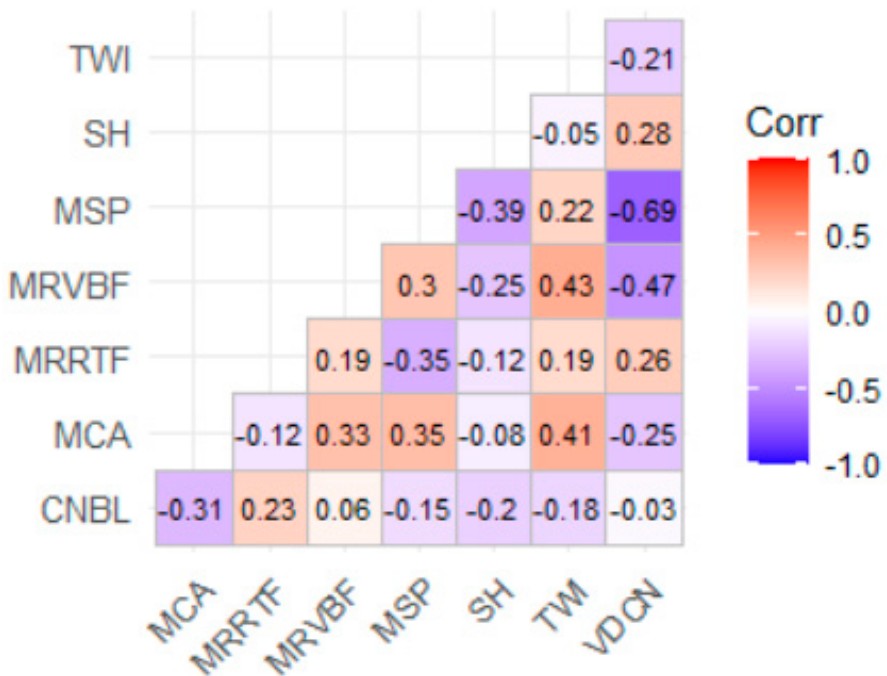

**Figure 4.** Correlation among the predictors.

*3.2. Base Learner Performances*

The prediction performance assessments for each base learner are summarized in Table 4. The average CCC values of the base learners ranged from 0.27 to 0.77 for sand content, 0.26 to 0.74 for silt content, 0.18 to 0.76 for clay content, 0.31 to 0.35 for SOC, 0.37 to 0.55 for pH, and 0.30 to 0.60 for topsoil depth; RMSE ranged from 5.07 to 10.79% for sand content, 4.99 to 8.89% for silt content, 1.85 to 3.72% for clay content, 0.73 to 0.76 g/kg for SOC, 0.32 to 0.50 for pH, and 5.38 to 9.27 cm for topsoil depth. Our results indicated that the RF model predicts well in all the soil properties. However, the GLM model had the poorest performances in all the soil properties, with a RMSE of 10.86% for sand content, 8.98% for silt content, 3.49% for clay content, 0.76 g/kg for SOC, 0.50 for pH, and 9.27 cm for topsoil depth. Although the standard deviations of these performance estimates show that there was substantial variation across cross-validation repetitions, it is evident that the observed differences in performance estimates are mostly substantial, relative to the random variability.

**Table 4.** Performance of base learners to predict soil properties based on 20 repeats, ten-fold cross validation.

| Soil Properties | Learners | CCC | | RMSE | |
|---|---|---|---|---|---|
| | | Mean | SD | Mean | SD |
| Sand | Cubist | 0.65 | 0.20 | 7.46 | 1.54 |
| | GLM | 0.27 | 0.20 | 10.79 | 2.31 |
| | GBM | 0.50 | 0.18 | 9.12 | 1.79 |
| | RF | **0.77** | **0.04** | **5.07** | **1.04** |
| | SVM | 0.47 | 0.23 | 9.56 | 2.21 |
| Silt | Cubist | 0.61 | 0.21 | 6.21 | 1.32 |
| | GLM | 0.26 | 0.22 | 8.89 | 2.01 |
| | GBM | 0.41 | 0.22 | 7.85 | 1.70 |
| | RF | **0.74** | **0.07** | **4.99** | **0.96** |
| | SVM | 0.31 | 0.22 | 8.45 | 1.74 |

**Table 4.** *Cont.*

| Soil Properties | Learners | CCC | | RMSE | |
|---|---|---|---|---|---|
| | | **Mean** | **SD** | **Mean** | **SD** |
| Clay | Cubist | 0.61 | 0.12 | 2.52 | 0.58 |
| | GLM | 0.18 | 0.14 | 3.72 | 0.48 |
| | GBM | 0.32 | 0.07 | 3.39 | 0.41 |
| | RF | **0.76** | **0.08** | **1.85** | **0.53** |
| | SVM | 0.54 | 0.19 | 2.71 | 0.61 |
| SOC | Cubist | 0.35 | 0.13 | 0.74 | 0.26 |
| | GLM | 0.31 | 0.13 | 0.76 | 0.29 |
| | GBM | 0.33 | 0.15 | 0.75 | 0.30 |
| | RF | **0.34** | **0.13** | **0.73** | **0.29** |
| | SVM | 0.32 | 0.12 | 0.73 | 0.28 |
| pH | Cubist | 0.59 | 0.22 | 0.42 | 0.12 |
| | GLM | 0.42 | 0.15 | 0.50 | 0.12 |
| | GBM | 0.40 | 0.20 | 0.50 | 0.11 |
| | RF | **0.55** | **0.06** | **0.32** | **0.07** |
| | SVM | 0.37 | 0.21 | 0.50 | 0.13 |
| Topsoil depth | Cubist | 0.60 | 0.18 | 7.45 | 2.02 |
| | GLM | 0.49 | 0.22 | 9.27 | 2.12 |
| | GBM | 0.30 | 0.19 | 8.59 | 2.18 |
| | RF | **0.60** | **0.10** | **5.38** | **1.28** |
| | SVM | 0.50 | 0.26 | 8.03 | 2.43 |

Note: the best-performing models are printed in bold, SD is standard deviation.

### 3.3. Stacked Ensemble Performances

The results of the two stacking approaches (Stack_GLM and Stack_GBM) for the prediction of the six soil properties are presented in Table 5. The GBM stacking model (Stack_GBM) achieves nominally better predictive performance than the GLM stacking model (Stacking_GLM) for sand, silt, and pH, while the GLM stacking model performs better for clay, SOC, and topsoil depth. Nevertheless, the standard deviation values indicate that performances show substantial variation and are statistically indistinguishable. Overall, the RF model exhibited the best performance and performed better than or equal to the stacking approaches.

**Table 5.** Ensemble model performance based on repeated ten-fold cross-validation.

| Soil Properties | Ensemble Models | CCC | | RMSE | |
|---|---|---|---|---|---|
| | | Mean | SD | Mean | SD |
| Sand | Stack_GLM | 0.42 | 0.22 | 11.43 | 2.59 |
| | Stack_GBM | **0.55** | **0.13** | **8.94** | **1.48** |
| Silt | Stack_GLM | 0.04 | 0.15 | 10.52 | 2.45 |
| | Stack_GBM | **0.33** | **0.15** | **7.98** | **1.25** |
| Clay | Stack_GLM | **0.55** | **0.13** | **2.42** | **0.50** |
| | Stack_GBM | 0.57 | 0.14 | 2.50 | 0.60 |
| SOC | Stack_GLM | 0.34 | 0.17 | 0.75 | 0.28 |
| | Stack_GBM | **0.34** | **0.16** | **0.73** | **0.29** |
| pH | Stack_GLM | 0.25 | 0.24 | 0.52 | 0.15 |
| | Stack_GBM | **0.32** | **0.20** | **0.51** | **0.14** |
| Topsoil | Stack_GLM | **0.50** | **0.17** | **7.02** | **1.88** |
| | Stack_GBM | 0.50 | 0.17 | 7.92 | 1.94 |

Note: the best-performing models are printed in bold.

### 3.4. Variable Importance

Figure 5a–f shows the set of environmental variables, used in the prediction of each soil property, in terms of their permutation-based importance, with respect to the RMSE. The most effective variables in the particle size distribution models (sand, silt, and clay

content) were VDCN and CNBL, while LULC is the most important variable in predicting topsoil depth, soil pH, and SOC content.

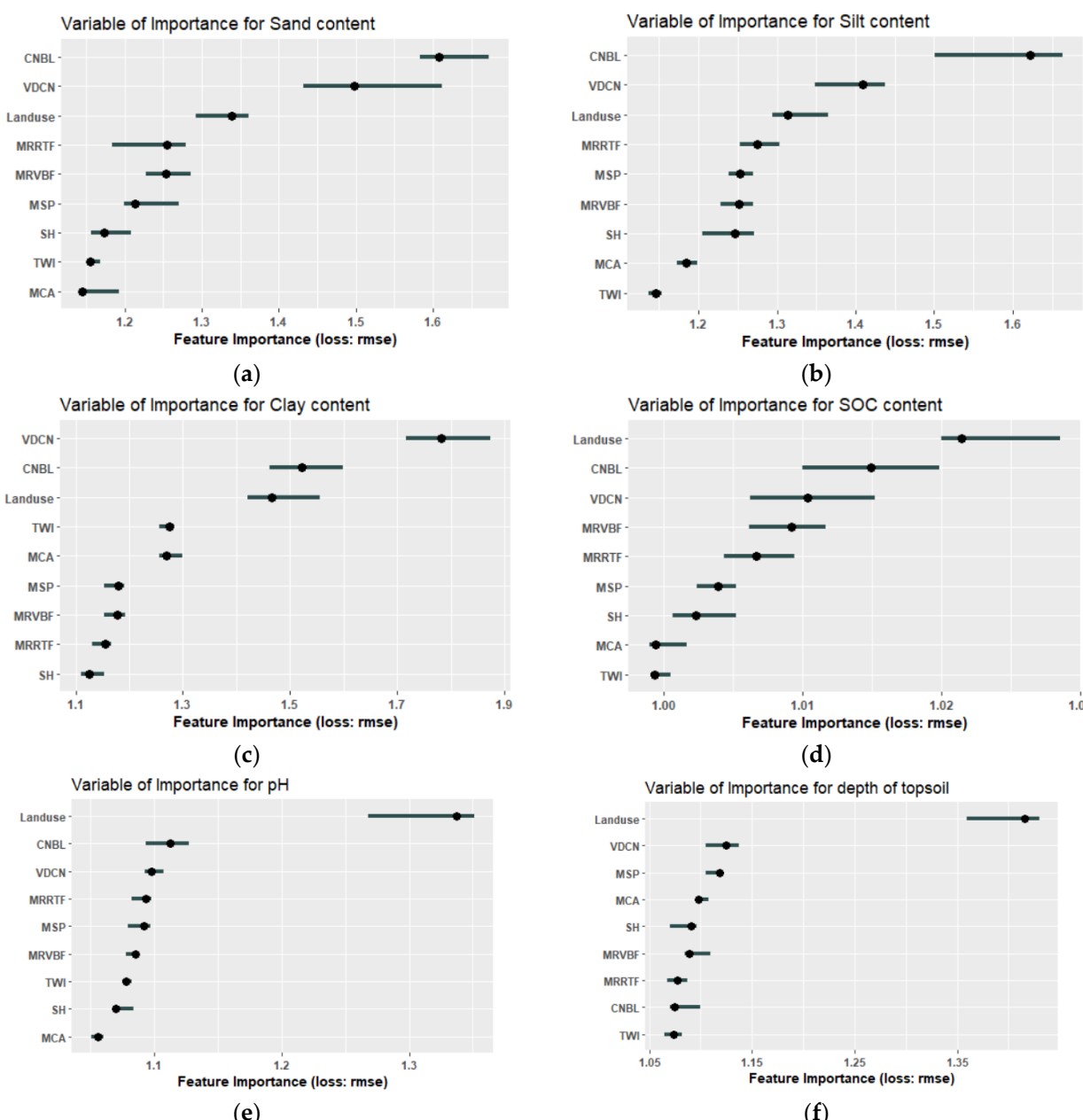

**Figure 5.** (**a**–**f**) Variable importance for different soil parameters derived by the best performing model.

### 3.5. Spatial Distribution of Soil Properties

The spatial distribution of all six soil properties, using the best-performing models, is depicted in Figure 6a–f. Low sand contents were predicted at high terrain units and high sand content at low terrain elevation. Moreover, there is a low clay content in low terrain units and low silt content at lower elevations, but silt and clay were predicted as being evenly distributed at higher terrace levels. The soil pH values were spatially predicted to be low on lower elevations and high on higher terrain units. Additionally, the topsoil depth and SOC content were spatially predicted, with low SOC content at higher terrace levels and high SOC content at lower terrain units.

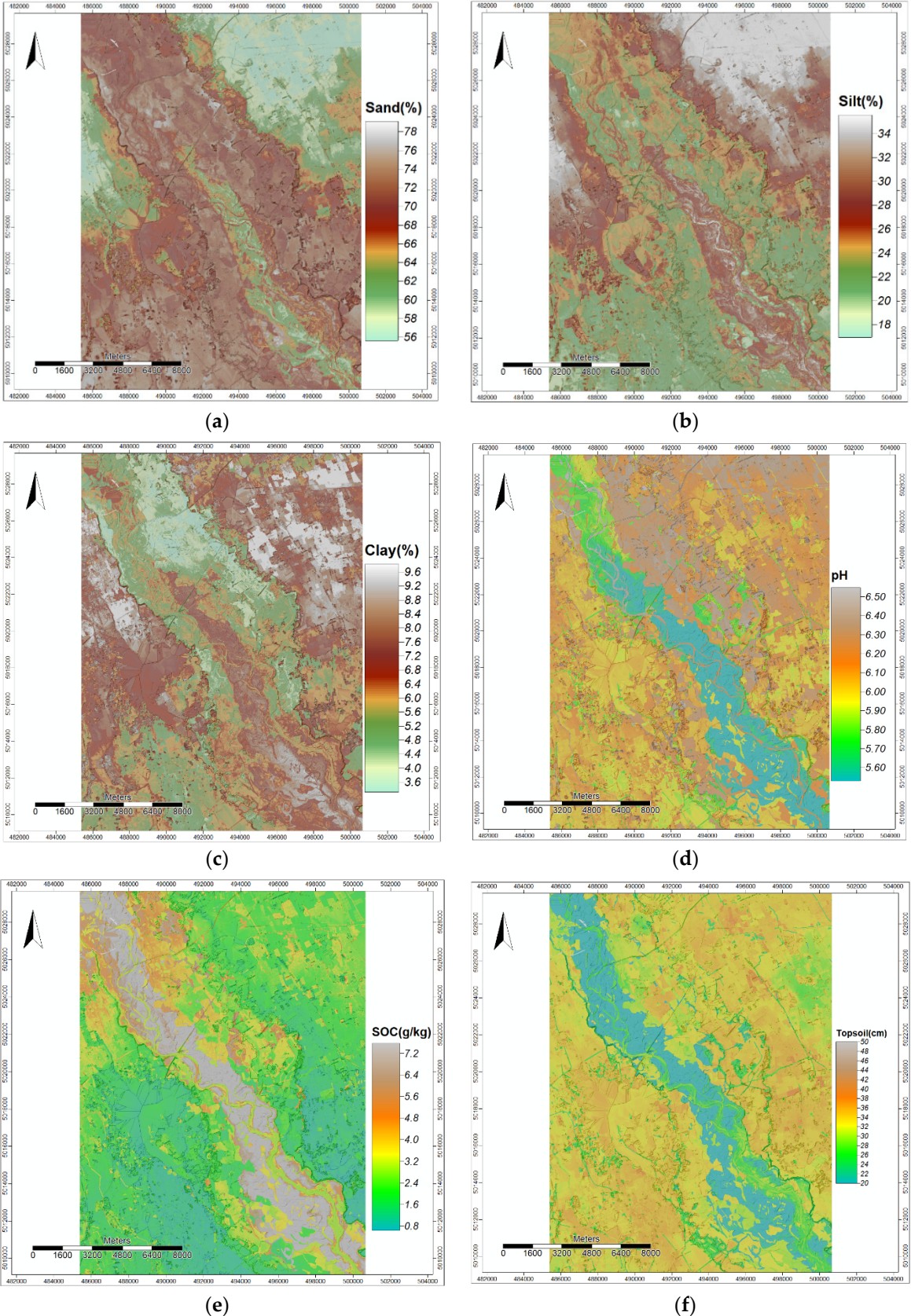

**Figure 6.** (**a**–**f**) Soil properties predicted with the best performing model for each response variable.

## 4. Discussion

Cubist, GBM, and RF are popular ensemble models used in DSM, all of which are based on regression trees. In this study, the RF model, as a bagging ensemble model, performed better than or at least equal to the Cubist and GBM models, based on the comparison of two statistical indicators (CCC and RMSE). This suggests that RF provides an excellent trade-off between model flexibility and the ability to avoid overfitting by tuning the hyperparameters [56]. The built-in sub-sampling of predictor variables also provides some protection against an over-reliance on a specific variable. Several studies have reported low RMSE for soil properties, developed by RF models, compared to other ML models [16,64–66]. Moreover, in Taghizadeh-Mehrjardi et al. [37], RF was indicated to be the best base learner among the 12 models used. However, RF models often vary significantly from study to study, and no single algorithm is 'best' within DSM and for every study area [12,17]. In addition, in our study, these three tree-based models mostly performed better than SVM and GLM. Though SVM can model nonlinear relationships, its performance is still susceptible to overfitting, and seeking optimal hyperparameters can be highly unstable. The GLM exhibited a poor performance in this study area because it cannot deal with the nonlinear relationships between the soil properties and environmental variables. Previous studies also showed that, when comparing both linear and nonlinear models, the tree-based learners are more effective than linear models [38,66].

The predictions from five individual models with different principles were combined using two stacking approaches: GLM and GBM. Neither of these two approaches were generally superior to the other one, considering variability in cross-validated performance estimates. However, in this study, the stacking models, in comparison to the base learners, seem to lag behind RF. This contradicted our original expectations based on previous studies [37,38,67,68]. In the study of Taghizadeh-Mehrjardi et al. [37], the super learner showed an improved performance in comparison to linear regression approaches by decreasing the RMSE by 46% on average. However, our results are similar to Zhang et al. [43], where nine models were used to construct an ensemble learner, using a super learner (SL) as a meta-learner to map soil pH for the Thompson-Okanagan region of British Columbia, and their overall finding was that the SL did not outperform all the other base learners. Moreover, Dobarco et al. [32] found that the ensemble predictions did not improve for silt and sand content but improved for clay content in their study.

We suggest that the non-superiority of the stacked models could be explained by the fact that the base learners are highly correlated (Appendix A Table A1). Moreover, stacked-model performance may depend on the quality of input datasets and the diversification of the input models [69]. An available literature review revealed that researchers often employed different methods or models in DSM, depending on the circumstances. Almost all of them stated that each model has its unique performance profile and specific strengths and weaknesses [12]. This uniqueness is mainly related to the complex nature and distinct mathematics of each model. Therefore, a comprehensive comparison of machine learning models for base learners and meta-learners is advisable, in order to check if the model outputs will yield substantially different results, before applying ensemble machine learning techniques as a means for improving predictions. Similarly, there might be an improvement in the performance if the ensemble model's residuals are spatially interpolated and then added to the deterministic spatial trend in the form of a regression kriging model. In addition, other studies have shown that each model could be strongly affected and improved by an increasing number of soil samples and additional environmental variables derived from remote sensing data or parent materials [70,71]. In our further studies, we will consider leveraging additional environmental variables to represent vegetation patterns and parent materials in the study area.

Mapping soil properties in an agricultural lowland area can be a challenge since soil forming factors, such as topography and vegetation, may not substantially correlate with soil properties, in space, to an extent at which they can be incorporated effectively in DSM [72]. However, terrain attributes, derived from high-resolution elevation data,

can capture local spatial variation that resulted from the interaction of water flows and topography [73]. Among the terrain attributes used in this study, VDCN and CNBL had highly significant correlations with all the soil properties and were ranked among the most influential variables. A similar trend was observed in a study presented by Kokulan et al. [74], where VDCN reflected the relationships between texture and erosion, and in Zhang et al. [39], where pH values were significantly correlated with CNBL and elevation. Both VDCN and CNBL are calculated from the drainage network, and they give information on the hydraulic gradients, in turn triggering soil erosion, as well as lateral and ground water fluxes [75]. Moreover, they facilitate the redistribution of fine material in this study area. However, since we are in a fluvial landscape, VDCN also reflects the age of the soils. Generally, higher elevations represent older terrace levels and hence, are characterized by mature and deep soils. Instead, the areas close to the river network are much younger, and thus, show only rudimentary and shallow soils. Concerning SOC, pH, and topsoil depth, land use seems to be the most important variable (Figure 5). This agrees with Adhikari et al. [76] who showed that land use was identified as one of the important variables that are related to SOC distribution at five standard soil depths. This can be explained by the direct relationship of land use and SOC in terms of plant cover and plant residues released to the soil. SOC content, predicted by the RF models, is generally higher on the lower terrace levels mainly covered by woodlands (forest and bushlands). Despite the distribution of agricultural areas and woodlands that show distinct differences in the SOC and pH, there are also differences in the agricultural areas themselves. In turn, they reflect the spatial distribution of certain crops like rice fields, simple arable lands, stable meadows, and permanent crops, as well as their respective irrigation schemes. Specific crops and/or vegetation need a certain top and subsoil water budget. These plants are influenced by their root system pH vales or SOC contents that, in turn, facilitate nutrient uptake. Particularly, lower pH is predicted in woodlands, whereas, on average, higher pH is modelled for arable land, while accounting for the other variables in the RF model. The latter might be due to carbonate applications by farmers. Moreover, vegetation directly affects pH by their residues and chemistry. Finally, in a lowland agricultural area, there might be changes in topography due to intensive agricultural activities; thus, using terrain attributes instead of absolute elevation can effectively explain soil patterns. However, it is striking that the predicted spatial distribution of SOC, pH, topsoil depth, and the soil texture classes, is illustrating the general distribution pattern related to the fluvial terrace levels and the vegetation, land use, and management.

## 5. Conclusions

In this study, linear and nonlinear machine learning models were applied to build a reliable and accurate estimation model to provide the spatial distribution of particle size distribution (sand, silt, and clay content), SOC content, pH, and the topsoil depth in an agricultural lowland area of Lombardy region, Italy, using terrain attributes and land use information. The nonlinear machine learning models generally show a good performance compared to the linear models. Overall, out of the five individual machine learning methods, RF performed best in this study. However, if RF and the other base learners are compared to the stacked ensemble models, none of these meta-learners stood out with superior performances. This suggests that a comprehensive comparison of machine learning models, for base learners and meta-learners, is advisable in order to check if the model outputs will yield substantially different results before applying ensemble machine learning techniques as a means for improving predictions.

In this study, we documented that, among the terrain attributes, CNBL and VDCN are the most important predictor variables explaining differences in soil properties in the study area. VDCN is related to the river terrace levels and, hence, to soil evolution stages, resulting in different soil depth, texture composition, and SOC content. However, land use and, particularly, crops are also related to the soil (pH, SOC, and topsoil depth) or reflect certain soil properties, like water availability and soil porosity. Furthermore, we

show that DSM, using ML models, has a high potential to effectively predict the spatial properties of soil attributes in lowland areas. We expect that further improvements in model accuracy could be achieved by incorporating additional environmental variables that represent vegetation patterns or the mineralogical composition of the topsoil.

**Author Contributions:** Conceptualization, O.D.A. and M.M.; methodology: O.D.A.; software: O.D.A.; validation: A.B. (Alexander Brenning), M.M. and O.D.A.; formal analysis: O.D.A.; investigation, A.B. (Alexander Brenning) and M.M.; resources: S.B. and A.B. (Alice Bernini); data curation, S.B., O.D.A. and A.B. (Alice Bernini); writing: O.D.A. and M.M.; writing review and editing: A.B. (Alexander Brenning) and M.M.; visualization: O.D.A.; supervision: A.B. (Alexander Brenning) and M.M.; project administration: M.M.; funding acquisition: M.M. All authors have read and agreed to the published version of the manuscript.

**Funding:** This research was supported by Regione Lombardia, POR FESR 2014-2020—Call HUB Ricerca e Innovazione, Project 1139857 CE4WE: Approvvigionamento energetico e gestione della risorsa idrica nell'ottica dell'Economia Circolare (Circular Economy for Water and Energy).

**Acknowledgments:** We acknowledge the support by ERSAF for providing soil data and the University of Pavia for supporting us with computing and laboratory facilities, as well as Deutsches Zentrum für Luft- und Raumfahrt (DLR) for supporting us with TanDEM-X data (grant DEM_Hydro1679). Furthermore, ERASMUS funds (N. 205/E2021-22) and University of Pavia (2021-UNPVCLE-0179492) sustained Odunayo Adeniyi during his training stages at Jena University fostering the EC2U cooperation.

**Conflicts of Interest:** The authors declare no conflict of interest.

## Appendix A

**Table A1.** Correlation among the predictions of the base learners.

|  |  | Cubist | GLM | GBM | RF | SVM |
|---|---|---|---|---|---|---|
| Sand | Cubist | 1.00 | 0.81 | 0.86 | 0.87 | 0.86 |
|  | GLM | 0.86 | 0.81 | 1.00 | 0.87 | 0.86 |
|  | GBM | 0.81 | 1.00 | 0.81 | 0.77 | 0.80 |
|  | RF | 0.87 | 0.77 | 0.87 | 1.00 | 0.88 |
|  | SVM | 0.86 | 0.80 | 0.86 | 0.88 | 1.00 |
| Silt | Cubist | 1.00 | 0.84 | 0.89 | 0.89 | 0.91 |
|  | GLM | 0.84 | 1.00 | 0.85 | 0.77 | 0.82 |
|  | GBM | 0.89 | 0.85 | 1.00 | 0.91 | 0.87 |
|  | RF | 0.89 | 0.77 | 0.91 | 1.00 | 0.89 |
|  | SVM | 0.91 | 0.82 | 0.87 | 0.89 | 1.00 |
| Clay | Cubist | 1.00 | 0.82 | 0.80 | 0.82 | 0.89 |
|  | GLM | 0.82 | 1.00 | 0.82 | 0.72 | 0.77 |
|  | GBM | 0.80 | 0.82 | 1.00 | 0.82 | 0.80 |
|  | RF | 0.82 | 0.72 | 0.82 | 1.00 | 0.82 |
|  | SVM | 0.89 | 0.77 | 0.80 | 0.82 | 1.00 |
| SOC | Cubist | 1.00 | 0.77 | 0.78 | 0.85 | 0.72 |
|  | GLM | 0.77 | 1.00 | 0.86 | 0.85 | 0.84 |
|  | GBM | 0.78 | 0.86 | 1.00 | 0.91 | 0.82 |
|  | RF | 0.85 | 0.85 | 0.91 | 1.00 | 0.80 |
|  | SVM | 0.72 | 0.84 | 0.82 | 0.80 | 1.00 |
| pH | Cubist | 1.00 | 0.76 | 0.77 | 0.83 | 0.81 |
|  | GLM | 0.76 | 1.00 | 0.85 | 0.70 | 0.79 |
|  | GBM | 0.77 | 0.85 | 1.00 | 0.82 | 0.79 |
|  | RF | 0.83 | 0.70 | 0.82 | 1.00 | 0.74 |
|  | SVM | 0.81 | 0.79 | 0.79 | 0.74 | 1.00 |
| Topsoil depth | Cubist | 1.00 | 0.73 | 0.79 | 0.84 | 0.81 |
|  | GLM | 0.73 | 1.00 | 0.83 | 0.68 | 0.74 |
|  | GBM | 0.79 | 0.83 | 1.00 | 0.82 | 0.82 |
|  | RF | 0.84 | 0.68 | 0.82 | 1.00 | 0.80 |
|  | SVM | 0.81 | 0.74 | 0.82 | 0.80 | 1.00 |

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
