# Peer review of "Digital Mapping of Soil Properties Using Ensemble Machine Learning Approaches in an Agricultural Lowland Area of Lombardy, Italy"

_land, doi:10.3390/land12020494_

Round 1
Reviewer 1 Report
The manuscript entitled "Digital mapping of soil properties using ensemble machine learning approaches in an agricultural lowland area of Lombardy, Italy" aims to evaluate and compare a stacked integration method with five machine learning models for predicting the content of some soil properties and mapping their spatial distribution. A lot of work has been done.
My major concerns:
This manuscript attempts to combine five different machine learning methods, three ensemble modeling methods, and eight topographic factors to predict and map six soil attributes. Nevertheless, the scientific contribution of this manuscript is not very clear. The abstract does not highlight the key points and the conclusions are vague and non-specific. The introduction is not focused on the research question. The results are put together with the discussion section, but the discussion cannot clearly explain the reasons behind the results. For example, the relationship between the unsatisfactory results and the data itself (e.g., data distribution), since the actual content of several soil properties is very low. Of course, other reasons should be considered as well. In terms of the results, even if the data are significantly correlated but the value of the correlation is not high (e.g., Table 3), then its explanatory power is questionable. If a separate machine learning algorithm yields more desirable results and ensemble modeling yields similar or worse results, why use ensemble modeling? You don't explain it clearly in the text. In addition, when performing stacked ensemble modeling, why only GLM and GBM are directly chosen instead of the other three machine learning algorithms as the base learners? The prediction maps produced for the 6 soil properties you also did not explore and explain in depth. The conclusions achieved need further investigation.
Author Response
Point 1: This manuscript attempts to combine five different machine learning methods, three ensemble modeling methods, and eight topographic factors to predict and map six soil attributes. Nevertheless, the scientific contribution of this manuscript is not very clear.
Response 1: We thank the reviewer for pointing this out. The main objective of this study is to evaluate and compare stacking ensemble approach with five ML models (base learners) to predict and map the spatial distribution of different soil properties such as texture (sand, silt, clay content), soil organic carbon (SOC), pH, and topsoil depth in an agricultural lowland area of Lombardy region, Italy. (138-145)
Point 2: The abstract does not highlight the key points and the conclusions are vague and non-specific.
Response 2: We thank the reviewer for pointing out this issue. The abstract is stated to be 200 words. Here’s the abstract:
A sustainable agricultural landscape management needs reliable and accurate soil maps and updated geospatial soil information. Recently, machine learning (ML) models have commonly been used in digital soil mapping together with limited data for various types of landscapes. In this study, we tested linear and non-linear ML models in predicting and mapping soil properties in an agricultural lowland landscape of Lombardy region, Italy. We further evaluate the ability of an ensemble learning model based on a stacking ap-proach to predict the spatial variation of soil properties such as sand, silt, and clay con-tents, soil organic carbon content, pH, and topsoil depth. Therefore, we by combined the predictions of the base learners (ML models) with two meta-learners. Prediction accura-cies were assessed using repeated ten-fold cross validation, which showed that the stack-ing models did not out-performed all the individual base learners for all the soil properties. The most important topographic predictors of the soil properties were vertical distance to channel network and channels distance to network base level. The results yield valuable information for a sustainable land use in an area with a particular soil water cycle as well as for future climate and socioeconomic changes influencing water content, soil pollution dynamics and food security. (29-44)
Point 3: The introduction is not focused on the research question.
Response 3: We thank the reviewer for pointing out this issue. The introduction has been modified and restructured (49-145)
Point 4: The results are put together with the discussion section, but the discussion cannot clearly explain the reasons behind the results.
Response 4: We thank the reviewer for pointing out this issue.The results have been separated now from the discusion.
Point 5: If a separate machine learning algorithm yields more desirable results and ensemble modeling yields similar or worse results, why use ensemble modeling? You don't explain it clearly in the text.
Response 5: We thank the reviewer for pointing out this issue. We attempted using ensembling modelling believing it will improve the accuracy of our base learners. (262)
Point 6: In addition, when performing stacked ensemble modeling, why only GLM and GBM are directly chosen instead of the other three machine learning algorithms as the base learners?
Response 6: We thank the reviewer for pointing out this issue. We tend to compare linear and non linear model as meta-learner. This has been added to the manuscript. (263-268)
Point 7: The prediction maps produced for the 6 soil properties you also did not explore and explain in depth. The conclusions achieved need further investigation.
Response 7: We appreciate this suggestion. We have modified the manuscript.

Reviewer 2 Report
The authors tested linear and non-linear machine learning models in predicting and mapping soil properties in an agricultural lowland landscape of Lombardy region, Italy. While the topic is interesting and a method for this purpose highly useful for digital soil mapping, the paper suffers from several contents that need to be thoroughly revised. Following, I provide some major suggestions for improving the paper.
1. Some statistical indexes of the different soil properties in Table 2 need to be added, for example, the skewness, to judge the distribution of data. This is the basis for selecting correlation analysis methods.
2. The model performances were evaluated using a cross validation method, and the authors selected ten-fold cross validation with three repetitions to optimize the model settings in this paper. The standard deviation of all data sets needs to be provided in detail.
3. Compared with Table 4 and Table 6, the performance of RF to predict soil properties was better than the stacked ensemble performances. For the ensemble machine learning approaches, random forest method was better? A lot of details and reasons should be explained clearly.
4. In the comparative analysis with existing studies, some quantitative analysis needs to be supplemented to avoid the unreliable results.
5. Line 320, the sentence was hard to understand, please revise it.
6. Result and discussion sections should be elaborated separately.
In addition, several styles need to be improved, such as annotation of multiple pictures, the references.
Author Response
Point 1: Some statistical indexes of the different soil properties in Table 2 need to be added, for example, the skewness, to judge the distribution of data. This is the basis for selecting correlation analysis methods
Response 1: We thank the reviewer for pointing out this issue. We agreed on this and the skewness has been added (313)
Point 2: The model performances were evaluated using a cross validation method, and the authors selected ten-fold cross validation with three repetitions to optimize the model settings in this paper. The standard deviation of all data sets needs to be provided in detail.
Response 2: We thank the reviewer for pointing out this issue. We agreed on this and the SD has been calculated and added too. Moreover, we increase the repetition from 3 to 20 in other to achieve a better stability of the model. (377)
Point 3: Compared with Table 4 and Table 6, the performance of RF to predict soil properties was better than the stacked ensemble performances. For the ensemble machine learning approaches, random forest method was better? A lot of details and reasons should be explained clearly.
Response 3: We thank the reviewer for pointing out this issue. We have modified this part.
Point 4: In the comparative analysis with existing studies, some quantitative analysis needs to be supplemented to avoid the unreliable results.
Response 4: Great point. We have modified the manuscript. (487)
Point 5: Line 320, the sentence was hard to understand, please revise it.
Response 5: We have modified the manuscript.
Point 6: Result and discussion sections should be elaborated separately.
Response 6: Great point. We have modified the manuscript.
Round 2
Reviewer 2 Report
Thanks to the authors to make extensive editing of the manuscript. Unfortunately, the author did not make a thorough revision according to my major suggestions.
In the correlation analysis section, as shown in Table 2, some soil parameters of the sampling sites have an abnormal distribution, for example, soil organic carbon. But the authors used Pearson's correlation analysis, as depicted in Table 3. It’s wrong. The inappropriate method used may lead to the uncertainty or unreliable results.
The authors attempted using stacking ensemble models to improve prediction accuracies of the soil properties. But the stacking models did not out-performed and the RF model performed better. Due to the lack of convincing explanations, the experimental results and the experimental design have some contradictions in this manuscript.
Author Response
Point 1: In the correlation analysis section, as shown in Table 2, some soil parameters of the sampling sites have an abnormal distribution, for example, soil organic carbon. But the authors used Pearson's correlation analysis, as depicted in Table 3. It’s wrong. The inappropriate method used may lead to the uncertainty or unreliable results.
Response 1: We thank the reviewer for pointing out this issue. Indeed outliers can have great influence on Pearson's correlations whereas univariate outliers do not exist with Spearman's rho because everything is converted to ranks. Thus, Spearman was adopted, and the Pearson’s correlation analysis done earlier was removed from the paper. However, the highly skewed variable (SOC) was log-transformed before modelling and back transformed during estimation of the predicted values.
Point 2: The authors attempted using stacking ensemble models to improve prediction accuracies of the soil properties. But the stacking models did not out-performed and the RF model performed better. Due to the lack of convincing explanations, the experimental results and the experimental design have some contradictions in this manuscript.
Response 2: We thank the reviewer for pointing out this issue. The objective of this study is to evaluate and compare a stacking ensemble model approach with five ML models (base learners) to predict and map the spatial distribution of different soil properties such as texture (sand, silt, clay content), soil organic carbon (SOC), pH, and topsoil depth in an agricultural lowland area of Lombardy region, Italy. Our result showed that the stacking models in comparison to the base learners were lag behind the RF which is also an ensemble model that uses bagging techniques. Thus contradicted our original expectations based on previous studies. Perhaps this performance could be explained by the fact that the base learners are highly correlated (Appendix Table 1). Moreover, the performance of stacking models may depend on the diversification of the input models and the quality of input datasets and . An available literature review revealed that researchers often employed different methods or models in DSM depending on the circumstances. Almost all of them stated that each model has its unique performance and has its strengths and weaknesses. This uniqueness is mainly related to the complex nature and distinct mathematics of each model. Therefore, one might possibly have an improved performance by changing the type and number of ML-models for base learners and meta-learning models. Therefore, comprehensive comparison of machine learning models for base learners and meta-learner is advisable in order to check if the model outputs will yield substantially different results before applying ensemble machine learning techniques as a means for improving predictions. Similarly, there might be an improvement in the performance if the ensemble model residuals are spatially interpolated and then added to the deterministic spatial trend in form of a regression kriging model. In addition, other studies have shown that each model could be strongly affected and improved by an increasing number of soil samples and additional environmental variables derived from remote sensing data or parent materials. In our further studies, we will consider leveraging on additional environmental variables to represent vegetative patterns and parent materials in the study area.